# Multimodal Assessment of Pancreatic Cancer Resectability Using Deep Learning

**Vincent Ochs**[1] (iD)                                          vincent.ochs@unibas.ch

**Christoph Kuemmerli**[2]                          Christoph.Kuemmerli@clarunis.ch

**Florentin Bieder**[1]                                       florentin.bieder@unibas.ch

**Julia Wolleb**[1]                                              julia.wolleb@unibas.ch

**Joël L. Lavanchy**[1,2] (iD)                          joel.lavanchy@clarunis.ch

**Julia Ruppel**[2]                                             Julia.Ruppel@clarunis.ch

**Jan Liechti**[2]                                                    j.liechti@unibas.ch

**Stephanie Taha-Mehlitz**[2]                       stephanie.taha@clarunis.ch

**Christian Andreas Nebiker**[3]                  Christian.Nebiker@ksa.ch

**Beat Müller**[2]                                           Beat.Mueller@clarunis.ch

**Giuseppe Kito Fusai**[4]                                      G.fusai@nhs.net

**Joerg-Matthias Pollok**[4]                    joerg-matthias.pollok@nhs.net

**Anas Taha**[1,5,6]                                               anas.taha@unibas.ch

**Philippe C. Cattin**[*1] (iD)                       philippe.cattin@unibas.ch

**Sebastian Staubli**[†2,4]                                    s.staubli@nhs.net

[1] *University of Basel, Department of Biomedical Engineering, 4123 Allschwil, Switzerland*

[2] *Clarunis, University Digestive Health Centre, Basel, Switzerland*

[3] *Department of General Surgery, Kantonsspital Aarau, Tellstrasse, 5000, Aarau, Switzerland*

[4] *HPB and Liver Transplant Service, Royal Free Hospital, London, United Kingdom*

[5] *Department of Visceral Surgery, Cantonal Hospital Basel-Land, Liestal, Switzerland*

[6] *Department of Surgery, East Carolina University, Brody School of Medicine, Greenville, North Carolina USA*

**Editors:** Accepted for publication at MIDL 2026

## Abstract

Accurate determination of pancreatic ductal adenocarcinoma (PDAC) resectability relies on evaluating how the tumor interacts with major peripancreatic vessels on CT imaging, yet expert assessment often shows substantial variability. We introduce a fully automated multimodal deep learning framework that jointly analyzes 3D contrast enhanced CT and structured clinical information to classify patients into the three National Comprehensive Cancer Network (NCCN) resectability categories (upfront resectable, borderline resectable, locally advanced). The approach uses a Swin-UNETR backbone to obtain anatomy aware image representations through auxiliary segmentation of pancreas, tumor, and vascular structures. These features are fused with a compact clinical embedding derived from 17 routinely collected variables and processed by a lightweight classification head. Model training is guided by a dynamic multitask objective that adapts the balance between segmentation and classification based on current tumor Dice performance, promoting feature representations that remain both anatomically informed and discriminative.

---

* Shared last authorship

† Shared last authorship

In a cohort of 159 patients (85 upfront resectable, 47 borderline resectable, 27 locally advanced), the proposed method achieved an AUC of 0.86, a macro-F1 of 0.79, and an accuracy of 0.85 using stratified nested 5-fold cross validation, outperforming adapted transformer based and geometric baseline approaches. External validation on an independent cohort with 52 patients from Kantonsspital Aarau (KSA Aarau) yielded an AUC of 0.86, a macro-F1 of 0.81, and an accuracy of 0.87, supporting cross-institution generalization.

Notably, the external KSA Aarau cohort contained complete clinical information for all variables used by the model and therefore did not require imputation. The comparable performance observed on this dataset suggests that the KNN based imputation applied to the training cohort did not introduce a detectable performance bias for the clinical variables considered.

Because segmentation labels are required only during training, the final system enables mask free inference while preserving vessel aware interpretability. These findings demonstrate that integrating anatomical supervision with clinical context yields a robust and reproducible tool for supporting operability (i.e., NCCN-based resectability) assessment in pancreatic cancer. The implementation is publicly available at `https://github.com/vincentochs/pancreas_resectability`, and the data, as well as the weights, can be made available by the corresponding author upon reasonable request.

**Keywords:** PDAC, Multi-Modality Model, Adaptive Loss Schedule, Resectability Classification.

## 1. Introduction

Pancreatic ductal adenocarcinoma (PDAC) is among the most lethal malignancies, accounting for over 90% of pancreatic cancers and exhibiting a 5 year survival rate below 10% (Sung et al., 2021; Siegel et al., 2023). Owing to its aggressive growth and the absence of specific early symptoms, pancreatic cancer is often diagnosed only at a late stage (Rawla et al., 2019). Complete surgical resection (R0) remains the only curative option, yet fewer than 20% of patients are eligible for surgery (Tempero et al., 2021; Neoptolemos et al., 2018).

Resectability is primarily determined by tumor involvement of key peripancreatic vessels, including the superior mesenteric artery (SMA), superior mesenteric vein (SMV), portal vein, and celiac trunk (Katz et al., 2013; Bockhorn et al., 2014). The NCCN defines three resectability categories, namely upfront resectable, borderline resectable, and locally advanced, based on these vascular relationships (Tempero et al., 2021). Accurate assessment on contrast enhanced computed tomography (CT) is critical for treatment planning (Isaji et al., 2018; Callery et al., 2009), yet inter-observer agreement remains poor, often below 70% even under standardized criteria (Giannone et al., 2021; Katz et al., 2013). As a result, operability assessment frequently relies on qualitative judgment rather than reproducible quantitative measurements. In this study, operability refers specifically to NCCN-based resectability categories, i.e., upfront resectable, borderline resectable, and locally advanced disease (Tempero et al., 2021).

Deep learning offers a path toward standardized CT interpretation by automatically extracting spatial and textural features (Litjens et al., 2017; Li et al., 2023). However, most prior works focus on segmentation (Lim et al., 2022; Zhou et al., 2020) or unimodal imaging based prediction (Huang et al., 2022), without integrating clinical variables such as comorbidities, tumor markers (CA19-9, CEA), and treatment history that influence decisions in multidisciplinary tumor boards (Hartwig et al., 2011; Rahib et al., 2014). Many

segmentation-based pipelines require segmentation masks at inference (manual or from an additional segmentation step), which can complicate clinical deployment and workflow integration (Bereska et al., 2024; Viviers et al., 2023).

To address these limitations, we propose an end-to-end multimodal framework that integrates 3D CT imaging with structured clinical data to predict NCCN resectability. By leveraging auxiliary anatomical supervision during training, the model learns vessel aware representations that generalize to unseen cases without requiring segmentation masks at test time, enabling a fully automated, interpretable decision support system for NCCN based resectability.

**Contributions.** The main contributions of this work are: (i) a clinically motivated multimodal framework integrating 3D CT imaging with structured clinical variables for NCCN resectability prediction; (ii) anatomy guided auxiliary supervision that enforces vessel aware feature learning without requiring segmentation masks at inference; (iii) a performance adaptive multitask objective that dynamically balances segmentation and classification throughout training; and (iv) a comprehensive evaluation including ablations and comparisons demonstrating improved predictive performance and clinical consistency over existing approaches.

## 2. Related Work

Automated resectability assessment in pancreatic cancer has largely focused on CT-based quantification of tumor vessel relationships. Bereska et al. used segmentation derived tumor–vessel contact lengths to approximate NCCN classes (Bereska et al., 2024), while Viviers et al. combined multiorgan and vessel segmentation with handcrafted geometric features for resectability prediction (Viviers et al., 2023).

Although effective, these pipelines require accurate segmentations at inference and do not incorporate patient level clinical information such as comorbidities or tumor markers. More recent approaches explore direct CT-based prediction of resectability or prognosis, but remain unimodal and often lack alignment with NCCN surgical frameworks (Schuurmans et al., 2025). For anatomical representation learning, convolutional encoder–decoders such as U-Net and its 3D variants are standard for abdominal organ and vessel segmentation (Ronneberger et al., 2015). Transformer based models like SwinUNETR further improve fine vascular delineation via shifted window self attention (Cao et al., 2021).

Pretraining on large abdominal datasets (e.g., BTCV, AMOS) enhances small structure generalization and accelerates downstream optimization (Ji et al., 2022). Multitask learning has also been explored, though most works rely on fixed loss-weighting between tasks and primarily emphasize anatomical accuracy rather than clinically actionable prediction (Kordnoori et al., 2024).

The integration of imaging with clinical variables is gaining traction in oncology. Early multimodal approaches fused radiomic or handcrafted imaging features with tabular data via late concatenation, offering limited cross-modal interaction (Jiang et al., 2021; Huang et al., 2016). More recent transformer-based models enable richer multimodal reasoning. For example, the Texture-Aware Transformer (TAT) captures texture variations across multiphase CT acquisitions for PDAC prognosis (Dong et al., 2023), while the Transformer-based

Multimodal Network for Segmentation and Survival prediction (TMSS) fuses CT, PET, and clinical data for head-and-neck cancer survival prediction (Saeed et al., 2022).

However, these works focus on longterm outcomes, not immediate surgical resectability, and generally omit explicit anatomical supervision. The present study builds on these foundations by combining anatomy guided representation learning with structured clinical fusion to deliver a fully automated NCCN-based resectability prediction model that is trained using NCCN based labels and anatomy guided supervision, without requiring manual segmentations at inference.

## 3. Method

We propose an end-to-end multimodal framework that predicts NCCN resectability from contrast enhanced CT and structured clinical data (Fig. 1). The model combines a 3D anatomy aware Swin-UNETR encoder–decoder with a tabular fusion head, trained jointly under a dynamic multitask objective. From each CT volume, the encoder produces a 256-dimensional anatomical feature vector while the decoder performs auxiliary multiorgan segmentation. In parallel, 17 clinical variables are processed through a multilayer perceptron (MLP) to yield a 32-dimensional clinical embedding. Both embeddings are concatenated into a 288-dimensional representation and passed to a classification head to predict the three NCCN resectability classes. During training, segmentation and classification gradients update the shared encoder, encouraging anatomically grounded and task relevant representations.

### 3.1. Data Modalities and Preprocessing

**CT Imaging:** Each volumetric CT scan is provided in NIfTI format and resampled to isotropic 1 mm spacing before being cropped or padded to $160 \times 160 \times 160$ voxels. This resolution preserves peripancreatic vessel continuity while keeping GPU memory usage within feasible limits. Voxel intensities are z-score normalized to account for acquisition variability. To improve robustness and generalization across scanners, we apply MONAI based augmentations including random flips, 90° rotations, affine transformations (rotation range $\pm 0.1$ rad), Gaussian noise ($\sigma = 0.01$), and random intensity shifts ($\pm 0.1$) (Cardoso et al., 2022).

The CT data were acquired at three institutions. To mitigate inter-site variability, all volumes were resampled to a common isotropic resolution, intensity-normalized, and processed using an identical preprocessing and augmentation pipeline. Clinical variables were harmonized using consistent definitions and units across institutions prior to model training.

**Segmentation Masks:** Voxelwise annotations were created and quality-controlled in a multi-expert workflow by up to five physicians. They comprise 16 anatomical and pathological classes: background, pancreas, tumor, portal vein, splenic vein, superior mesenteric vein (SMV), superior mesenteric artery (SMA), celiac trunk, aorta, inferior vena cava, common hepatic artery, proper hepatic artery, gastroduodenal artery, splenic artery, dilated pancreatic duct, and bile duct. The segmentation loss inherently emphasizes tumor and major

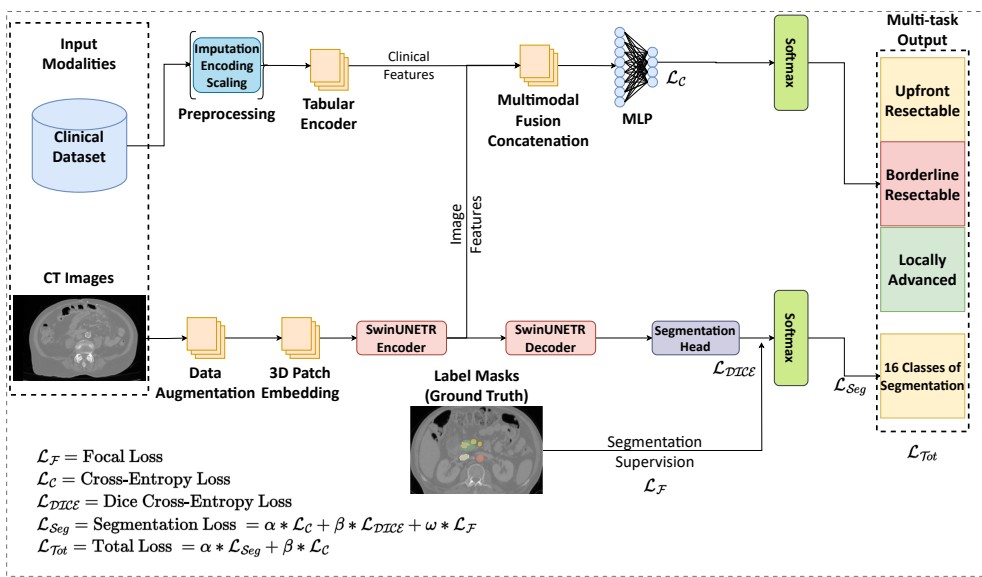

Figure 1: Proposed Network Architecture. During backpropagation, gradients from both the segmentation decoder and the classification fusion head are propagated through the shared encoder. This joint optimization enables the encoder to learn anatomically meaningful, discriminative features for resectability prediction.

vessels (SMA, SMV, portal vein, celiac trunk) because the Dice and Focal components give higher relative importance to small, clinically relevant structures.

Because the segmentation decoder and classification head share a common encoder, this emphasis encourages the encoder to learn vessel-aware anatomical features that are subsequently used for multimodal fusion and resectability classification. All masks were resampled to isotropic 1 mm spacing, cropped or padded to the $160 \times 160 \times 160$ field-of-view used for CT preprocessing, and aligned with the normalized CT volumes. These preprocessed masks serve as ground truth for auxiliary segmentation supervision during training.

**Clinical Data:** Seventeen structured variables capture complementary prognostic information, encompassing age at operation or diagnosis, sex, body mass index (BMI), Charlson Comorbidity Index (CCI), American Society of Anesthesiologists (ASA) score, diabetes status, nicotine use, alcohol consumption, CA 19-9 serum baseline level, CA 19-9 delta, CA 19-9 serum level after neoadjuvant therapy, CEA serum level, blood glucose, HbA1c, bilirubin at initial diagnosis, histopathological grade (G), and neoadjuvant (radio-)chemotherapy status.

The selected clinical variables were pre-defined in collaboration with experienced clinicians and reflect routinely available factors considered during multidisciplinary tumor board decision making rather than an exhaustive set of all disease-associated variables. Overall missingness across the 17 clinical variables was low (less than 5% of entries), with most variables being complete or near-complete. The highest missingness was observed for HbA1c (5%). Missing values were imputed hierarchically: KNN ($k = 5$) for continuous variables and mode imputation for categorical attributes. Categorical variables were one-hot encoded, and continuous features were z-score standardized (i.e., transformed to zero mean and unit variance based on the training split).

The processed inputs are passed through a three layer MLP (hidden dimensions: 64 and 32, each followed by LayerNorm and ReLU), producing a final 32-dimensional clinical embedding.

### 3.2. Model Architecture

**Swin-UNETR Backbone:** We adopted the Swin-UNETR (Cao et al., 2021) as the imaging encoder–decoder backbone. Its shifted window self-attention captures both local and global 3D contexts, crucial for fine vascular delineation around the pancreas. The encoder and decoder share skip connections for anatomical reconstruction, while the encoder bottleneck outputs a 256-dimensional feature vector used for both segmentation and classification.

This bottleneck representation constitutes a merged embedding of local and global features learned through Swin-UNETR's hierarchical, shifted-window self-attention.

Pretraining on the BTCV dataset provides anatomical priors and accelerates convergence.

**Multimodal Fusion:** To integrate imaging and clinical information, the 256-dimensional imaging feature is concatenated with the 32-dimensional clinical embedding to form a 288-dimensional multimodal vector. This fused representation is processed by an MLP fusion head (LayerNorm, ReLU, dropout) and a softmax classifier to predict the three NCCN-based resectability categories. Gradients from both segmentation and classification update

the shared encoder, enforcing anatomically informed features. We use simple concatenation for fusion, as ablation experiments showed no measurable benefit from more complex cross-attention mechanisms.

### 3.3. Multi-Task Learning Objective

The model was trained via a joint objective combining segmentation and classification tasks:

$$\mathcal{L}_{\text{total}} = w_{\text{seg}}(\tau)\,\mathcal{L}_{\text{seg}} + w_{\text{cls}}(\tau)\,\mathcal{L}_{\text{cls}}, \tag{1}$$

where $\tau$ denotes the tumor Dice score and $w_{\text{seg}}$ and $w_{\text{cls}}$ are the segmentation and classification loss weights, respectively. Both weights change dynamically as a function of the current segmentation performance $\tau$. This adaptive weighting gradually shifts emphasis from segmentation to classification as segmentation accuracy improves: early in training, when $\tau$ is low, a higher $w_{\text{seg}}$ emphasizes anatomical learning, whereas with improving segmentation quality the weight shifts toward classification by increasing $w_{\text{cls}}$. In the following, we discuss these components in more detail.

**Segmentation Loss:** The segmentation decoder is optimized using a hybrid loss that combines cross-entropy, Dice, and focal components:

$$\mathcal{L}_{\text{seg}} = \alpha\,\mathcal{L}_{\text{CE}} + \beta\,\mathcal{L}_{\text{Dice}} + \omega\,\mathcal{L}_{\text{Focal}}. \tag{2}$$

This encourages precise boundary delineation and robustness to class imbalance, particularly for small vessels.

**Classification Loss:** To account for the class imbalance in our dataset (85 upfront resectable, 47 borderline resectable, 27 locally advanced cases), we apply a class-weighted cross-entropy loss:

$$\mathcal{L}_{\text{cls}} = -\sum_{k=1}^{3} v_k\,y_k \log \hat{y}_k, \tag{3}$$

where $y = (y_1, y_2, y_3)$ denotes the one-hot encoded ground-truth class label, $\hat{y} = (\hat{y}_1, \hat{y}_2, \hat{y}_3)$ are the predicted class probabilities from the softmax output, and $v = [1.0, 2.0, 2.5]$ are smoothed inverse-frequency weights that increase the contribution of minority classes (borderline resectable, locally advanced) without overcompensating for their smaller sample sizes. These values were selected empirically to balance gradient magnitudes and ensure stable convergence across folds. A sensitivity analysis comparing different weighting schemes (e.g., strict inverse frequency and unweighted baselines) is presented in the appendix (see Table 4), demonstrating that the proposed configuration achieves the best trade-off between classification AUC and stability across runs.

**Adaptive Weighting:** The relative importance of segmentation and classification is adjusted dynamically according to the current segmentation performance:

$$w_{\text{seg}}(\tau) := \begin{cases} 3.0 & \tau \in [0, 0.1) \\ 1.5 & \tau \in [0.1, 0.5) \\ 1.0 & \tau \in [0.5, 1] \end{cases} \qquad w_{\text{cls}}(\tau) := \frac{1}{w_{\text{seg}}(\tau)}. \tag{4}$$

The thresholds reflect typical stages of network convergence observed during pilot training. Early epochs often yield tumor Dice scores below 0.1 due to unstable vessel localization, mid training stabilizes around 0.3-0.5 as anatomical structures emerge, and values above 0.5 indicate sufficiently reliable feature representations. The corresponding weights (3.0, 1.5, 1.0) were empirically chosen to maintain stable gradients, strongly emphasizing anatomical supervision when feature learning is unstable, then progressively shifting focus toward the classification objective as segmentation accuracy improves. This strategy mirrors a self paced or curriculum learning scheme, allowing the model to first learn anatomical context before optimizing the downstream NCCN-based resectability prediction. A detailed ablation of alternative threshold and weighting configurations, demonstrating the stability and performance benefits of the proposed schedule, is provided in the appendix (see Table 5).

## 4. Experiments and Results

### 4.1. Dataset

This retrospective multimodal cohort comprised 159 patients with histologically confirmed PDAC from the University Hospital Basel and St. Clara Hospital Basel. All data were de-identified and collected under institutional review board (IRB) approval. Ground-truth NCCN resectability labels were defined during routine clinical care by the multidisciplinary tumor board (MDT) at University Hospital Basel and St. Clara Hospital Basel, involving multiple physicians (e.g., surgery, radiology, oncology) according to NCCN criteria. To further ensure label consistency, MDT-based labels (upfront resectable, borderline resectable, locally advanced) were independently reviewed and validated by an additional physician who was not involved in model development. For external validation, we additionally collected an independent cohort of 52 patients from Kantonsspital Aarau (KSA Aarau), which was not used for model development or internal cross-validation.

According to NCCN criteria, the cohort included 85 upfront resectable (53.5%), 47 borderline resectable (29.5%), and 27 locally advanced cases (17%), each with corresponding contrast enhanced CT imaging and structured clinical data. The external dataset included 28 upfront resectable cases, 14 borderline resectable cases, and 10 locally advanced cases.

### 4.2. Experimental Setup

We employed a stratified nested 5-fold cross-validation scheme to ensure balanced representation of resectability categories and to obtain an unbiased performance estimate. For each outer fold, 20% of the dataset was held out as an independent test set and was never used during training, hyperparameter tuning, or early stopping. The remaining 80% served as the training pool for an inner split, where we partitioned the data into 64% training and 16% validation. This inner validation set was used exclusively for model selection, early stopping, and tuning of segmentation and classification losses. Training used the AdamW optimizer with cosine annealing scheduling, an initial learning rate of $10^{-4}$, weight decay of $10^{-5}$, and early stopping after 20 epochs without improvement on the inner validation set. The batch size was fixed at 16. Hyperparameters were optimized within the inner loop via grid search over learning rate $\{10^{-3}, 5{\cdot}10^{-4}, 10^{-4}\}$, feature size $\{24, 32, 48\}$, batch size $\{8, 16\}$, and segmentation loss coefficients $(\alpha, \beta, \omega) \in [0.2, 0.6]$. For each outer fold, the

configuration yielding the highest inner validation macro-F1 score was selected and subsequently evaluated on the held-out outer test set. Final performance metrics were averaged across all five outer folds.

All models were implemented in PyTorch using the MONAI framework and trained on an NVIDIA A100 (40GB) GPU. During inference, 3D CT volumes were processed using a sliding window strategy with Gaussian blending (Cardoso et al., 2022). Segmentation outputs were used only for evaluation, whereas the classification branch generated NCCN resectability predictions without requiring segmentation masks at test time.

### 4.3. Evaluation Metrics

Model performance was assessed for both segmentation and classification. Segmentation quality was measured using the Dice Similarity Coefficient (DSC), with emphasis on tumor and vessel classes relevant for surgical planning. Classification performance was evaluated using Accuracy, macro averaged F1-score, and the Area Under the ROC Curve (AUC), capturing overall discrimination and classwise balance across the three NCCN resectability categories.

### 4.4. NCCN Geometry Analysis

To assess whether the model implicitly learns NCCN related geometric patterns, we performed a post-hoc analysis of tumor vessel contact angles on the held out outer test sets. For each patient, predicted multiorgan segmentations were used to extract approximate arterial and venous contact angles around key vessels (arterial: SMA or celiac trunk; venous: SMV or portal vein). Although NCCN resectability definitions consider several factors such as encasement, deformity, occlusion, and reconstructability angle based surrogates provide a simplified approximation: arterial involvement of 1-180° often corresponds to borderline resectability, while $\geq 180°$ indicates locally advanced disease; similarly, venous contact $\geq 180°$ may suggest borderline involvement.

These surrogate angles were plotted in a 2D plane and compared with ground truth NCCN classes and model predictions. The observed clusters show that upfront resectable, borderline, and locally advanced cases tend to occupy the expected angles, suggesting that the model's anatomical representations capture geometric behavior despite the absence of angle-based supervision. This analysis was purely diagnostic and did not influence training, validation, or model selection. The visualization is shown in Figure 2 in the appendix.

### 4.5. Results

Although segmentation is used only as an auxiliary task, it plays a central role in shaping the anatomy aware encoder. The Swin-UNETR decoder produced stable multiorgan segmentations across folds, achieving DSC values of $0.82 \pm 0.03$ for pancreas, $0.71 \pm 0.05$ for tumor, and $0.67 \pm 0.06$ for major vessels. Representative qualitative examples illustrating pancreas, tumor, and major vessel segmentations are shown in Appendix Figure 3. The lower vessel performance reflects the intrinsic difficulty of segmenting small, low contrast vascular structures, yet even these imperfect masks provide sufficiently strong supervisory signals to regularize the encoder and improve downstream resectability prediction. This

demonstrates that although segmentation is an auxiliary task and not used at inference, the model does use the resulting latent anatomical features learned through the segmentation decoder; these features substantially improve the structure and clinical relevance of the imaging representations used by the classifier.

For the classification task, the proposed multimodal framework achieved strong and consistent performance across nested cross-validation folds, with an accuracy of 0.85, a macro-F1 score of 0.79, and an AUC of 0.86 (Table 1).

To assess generalization across institutions, we performed an external validation on an independent cohort from Kantonsspital Aarau (KSA Aarau). This dataset was not used during model development, hyperparameter tuning, or internal cross-validation. On this external cohort, the proposed method achieved an AUC of 0.86, a macro-F1 score of 0.81, and an accuracy of 0.87 (Table 2).

Despite differences in patient characteristics and institutional acquisition settings, the model maintained performance comparable to the internal cross-validation results, demonstrating robustness to inter-institutional domain shift. These findings support the generalizability of the proposed multimodal framework beyond the originating centers.

Class-wise F1-score variability across folds is reported in Appendix Table 6, showing stable performance despite the smaller sample size of the locally advanced class.

Table 1: Classification results. Comparison of different methods on stratified nested 5-fold cross-validation (CV). Results are reported as mean ± standard deviation across folds.

| Method | AUC | Macro-F1 | Accuracy |
|---|---|---|---|
| (Viviers et al., 2023) (Segmentation-Based) | 0.79 ± 0.04 | 0.74 ± 0.03 | 0.76 ± 0.04 |
| TAT (adapted from (Dong et al., 2023)) | 0.83 ± 0.03 | 0.77 ± 0.03 | 0.81 ± 0.03 |
| **Ours (Multimodal Swin-UNETR)** | **0.86 ± 0.03** | **0.79 ± 0.02** | **0.85 ± 0.03** |

Table 2: External validation results on an independent cohort from Kantonsspital Aarau (KSA Aarau).

| Method | AUC | Macro-F1 | Accuracy |
|---|---|---|---|
| (Viviers et al., 2023) (Segmentation-Based) | 0.82 | 0.77 | 0.81 |
| TAT (adapted from (Dong et al., 2023)) | 0.84 | 0.79 | 0.84 |
| **Ours (Multimodal Swin-UNETR)** | **0.86** | **0.81** | **0.87** |

## 4.6. Ablation Studies: Feature Isolation, Knockout Robustness, and Single-Modality Training

We conducted a series of ablation experiments to disentangle the contribution and robustness of each modality (Table 3). First, in the feature isolation setting, the pretrained Swin-UNETR encoder and tabular MLP were frozen, and new classification heads were trained

Table 3: Combined ablation study. FI = Feature Isolation, KO = Knockout, SM = Single-Modality.

| Type | Configuration | AUC | F1-Macro | Accuracy |
|------|---------------|-----|----------|----------|
| FI | Imaging (frozen) | 0.83±0.03 | 0.77±0.03 | 0.82±0.03 |
| | Tabular (frozen) | 0.75±0.04 | 0.72±0.03 | 0.73±0.04 |
| KO | Tabular → mean embedding | 0.83±0.03 | 0.76±0.03 | 0.81±0.03 |
| | Imaging → mean embedding | 0.77±0.04 | 0.70±0.04 | 0.73±0.04 |
| SM | CT-only | 0.82±0.03 | 0.76±0.03 | 0.78±0.03 |
| | Tabular-only | 0.74±0.04 | 0.71±0.04 | 0.73±0.04 |
| **Full multimodal (baseline)** | | **0.86±0.03** | **0.79±0.02** | **0.85±0.03** |

independently on their latent embeddings. This assessed the discriminative strength of each modality without re-training the fusion dynamics (FI block in Table 3).

Next, to evaluate resilience under partial input failure, a knockout analysis was performed in which one modality was replaced at inference by its mean embedding while the other remained intact. This quantified how strongly the fused classifier depends on each pathway and how well it tolerates missing or corrupted inputs (KO block).

Finally, single modality re-training was carried out by training CT only and tabular only models from scratch. These unimodal baselines revealed the inherent predictive capacity of each modality and highlighted the complementary nature of anatomical and clinical information in the multimodal model (SM block).

## 4.7. Comparisons with Existing Methods

To contextualize our framework, we re-implemented and adapted two representative approaches: (i) the Texture-Aware Transformer (TAT) by Dong *et al.* (Dong et al., 2023), originally developed for multiphase CT based survival prediction using texture-enhanced transformers and a neural distance module; and (ii) the segmentation-based assessment pipeline by Viviers *et al.* (Viviers et al., 2023), which computes handcrafted geometric features (e.g., tumor–vessel contact length, encasement, vessel involvement) from multiorgan segmentations to infer resectability according to the Dutch Pancreatic Cancer Group (DPCG) criteria. For comparability with our NCCN-based setting, we retained their geometric feature extraction but replaced the original DPCG rule based decision process with a supervised classifier trained to predict the three NCCN resectability categories.

Both baseline methods were adapted to match our single phase CT setting, incorporate structured clinical variables, and address the NCCN-based three class prediction task. Comparative results are summarized in Table 1.

TAT combines a texture encoder with two geometry focused components: a structural branch that processes local 3D patches around the shortest tumor–vessel distances, and a neural distance module that captures proximity between the tumor and major vessels (SMA, SMV, portal vein, celiac trunk). For our task, we adapted TAT to single phase CT, replaced its survival regression head with a softmax classifier, and retrained all geometry modules to learn NCCN style interactions directly from tumor–vessel surfaces. Training

schedules and resolutions were aligned with our framework for fairness. As shown in Table 1, our multimodal anatomy guided model outperformed the adapted TAT, largely due to the added clinical information and segmentation-based anatomical supervision.

For the Viviers-based baseline, the geometric descriptors extracted from our 16-class segmentations were used directly as input to the NCCN classifier. While this produced a strong handcrafted geometry baseline, our multimodal framework surpassed it by leveraging jointly learned imaging and clinical representations that capture anatomical and contextual patterns not easily encoded through handcrafted rules.

## 5. Conclusion

We presented an end-to-end multimodal framework for automated assessment of PDAC resectability that integrates 3D CT with structured clinical data. By combining anatomy aware segmentation supervision with a multimodal fusion head, the model learns vessel informed and clinically aligned representations for NCCN-based resectability prediction.

The method achieved strong classification performance across resectability categories, and ablation studies showed that imaging and clinical features provide complementary information: CT captures detailed tumor–vascular morphology, while clinical variables contribute essential patient specific context. Compared to unimodal and segmentation only baselines, the multimodal design consistently improved predictive accuracy.

Beyond performance, the framework introduces several methodological advances for this domain. Segmentation guidance encourages the encoder to learn vessel aware anatomical structure while avoiding the need for segmentation masks at inference time. The adaptive multitask objective stabilizes training by shifting focus from anatomical learning to classification as segmentation improves. Furthermore, the model represents the first end-to-end integration of 3D vessel aware CT features with structured clinical covariates specifically tailored to NCCN defined resectability assessment.

Limitations include the moderate cohort size and the use of semi-automated segmentation masks for auxiliary supervision.

Although clinical variables in the training cohort required imputation, no imputation was necessary for the external validation dataset. The comparable performance observed across cohorts suggests that the applied imputation strategy did not introduce a substantial bias in model predictions; however, further validation on additional external datasets needs to be conducted.

Although the reference standard reflects MDT consensus, we did not perform a dedicated multi-reader study comparing individual clinician performance with the model under standardized reading conditions. Such a prospective reader study (including assessment of interobserver agreement and comparison to MDT consensus) is an important next step for clinical deployment.

Explainability analyses and feature importance will be essential to assess generalization and clinical robustness.

More expressive fusion operators (e.g., cross-attention) may become beneficial at larger cohort sizes; in our experiments, they did not improve performance and reduced training stability, so we opted for the simplest reliable fusion for this dataset. Future work will involve multi-institutional evaluation, integration of additional imaging modalities such as

multiphase CT or histopathology, and exploration of more expressive fusion mechanisms. Overall, by jointly modeling anatomical detail and patient context, this framework offers a reproducible and clinically meaningful step toward objective, reliable decision support in pancreatic cancer surgery.

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

## Acknowledgments

This work was financially supported by the University of Basel through the Research Fund for Junior Researchers, which promotes the academic careers of outstanding junior researchers Sebastian M. Staubli was the recipient. In addition, Joël L. Lavanchy was funded by the Swiss National Science Foundation (P5R5PM 21766), the Novartis Foundation for medical-biological Research (23C162) and by the Vontobel Foundation (0867/2024).

## Appendix A. Ablation Studies

Table 4: Ablation of class weighting schemes for the nested cross-entropy loss (5-fold CV). The proposed smoothed weighting achieves the best balance between performance and stability.

| Scheme | Weights | AUC | Macro-F1 |
|---|---|---|---|
| Unweighted | (1, 1, 1) | 0.83±0.03 | 0.75±0.03 |
| Inverse-freq. | (1, 1.8, 3.2) | 0.85±0.03 | 0.77±0.03 |
| **Smoothed (Ours)** | **(1, 2, 2.5)** | **0.86±0.03** | **0.79±0.02** |

Table 5: Ablation study comparing different weighting strategies between segmentation and classification losses (Equation 1). Results were averaged across nested 5-fold cross-validation. The proposed Dice dependent adaptive weighting scheme achieves the best tradeoff between segmentation stability and classification accuracy.

| Weighting Strategy | AUC | Macro-F1 | Mean Tumor Dice |
|---|---|---|---|
| Fixed weights ($w_{\mathrm{seg}} = 1.0$, $w_{\mathrm{cls}} = 1.0$) | $0.84 \pm 0.03$ | $0.77 \pm 0.03$ | $0.49 \pm 0.05$ |
| Fixed weights ($w_{\mathrm{seg}} = 2.0$, $w_{\mathrm{cls}} = 0.5$) | $0.83 \pm 0.03$ | $0.76 \pm 0.03$ | $0.52 \pm 0.04$ |
| Fixed weights ($w_{\mathrm{seg}} = 0.5$, $w_{\mathrm{cls}} = 2.0$) | $0.80 \pm 0.04$ | $0.74 \pm 0.04$ | $0.42 \pm 0.06$ |
| **Adaptive (proposed)** | $\mathbf{0.86 \pm 0.03}$ | $\mathbf{0.79 \pm 0.02}$ | $\mathbf{0.56 \pm 0.04}$ |

Table 6: Per-class F1-scores across stratified nested 5-fold cross-validation. Results are reported as mean ± standard deviation across outer folds.

| NCCN Class | F1-score (mean ± SD) |
|---|---|
| Upfront resectable | $0.83 \pm 0.04$ |
| Borderline resectable | $0.78 \pm 0.06$ |
| Locally advanced | $0.70 \pm 0.07$ |

## Appendix B. NCCN Geometry Analysis

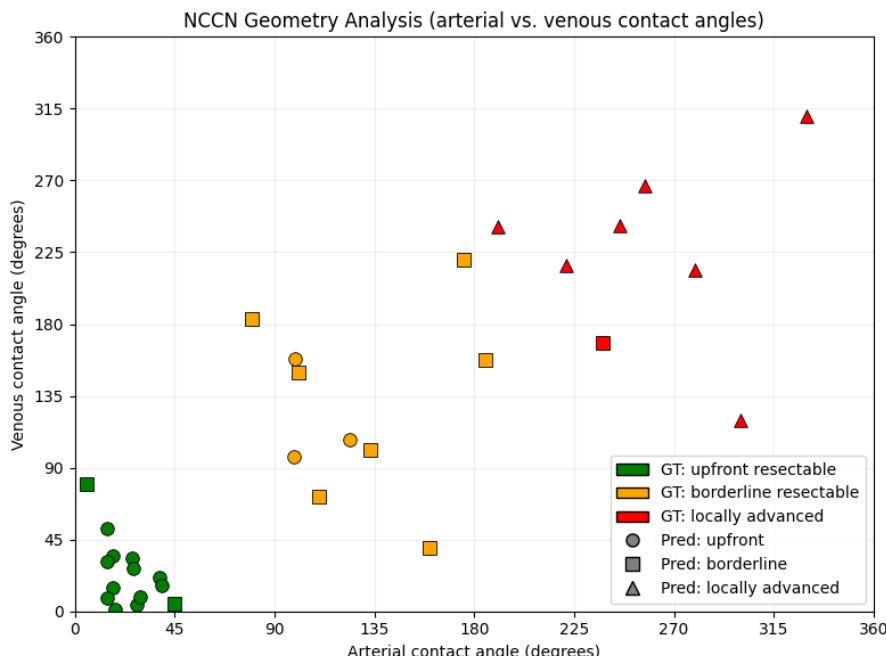

Figure 2: NCCN geometry analysis on the outer test folds. Arterial and venous tumor vessel contact angles were approximated from the model predicted segmentation masks and plotted against the ground truth NCCN resectability labels. Colors indicate the three NCCN classes (upfront resectable, borderline resectable, locally advanced), while marker shape denotes the model's predicted class. Because the original dataset does not contain explicit geometric measurements and NCCN resectability additionally depends on factors beyond contact angles (e.g., vessel occlusion and reconstructability), a perfect one-to-one correspondence between regions and ground-truth labels is not expected. The plot serves as a sanity check showing that the learned representations exhibit NCCN consistent geometric trends.

## Appendix C. Qualitative Examples

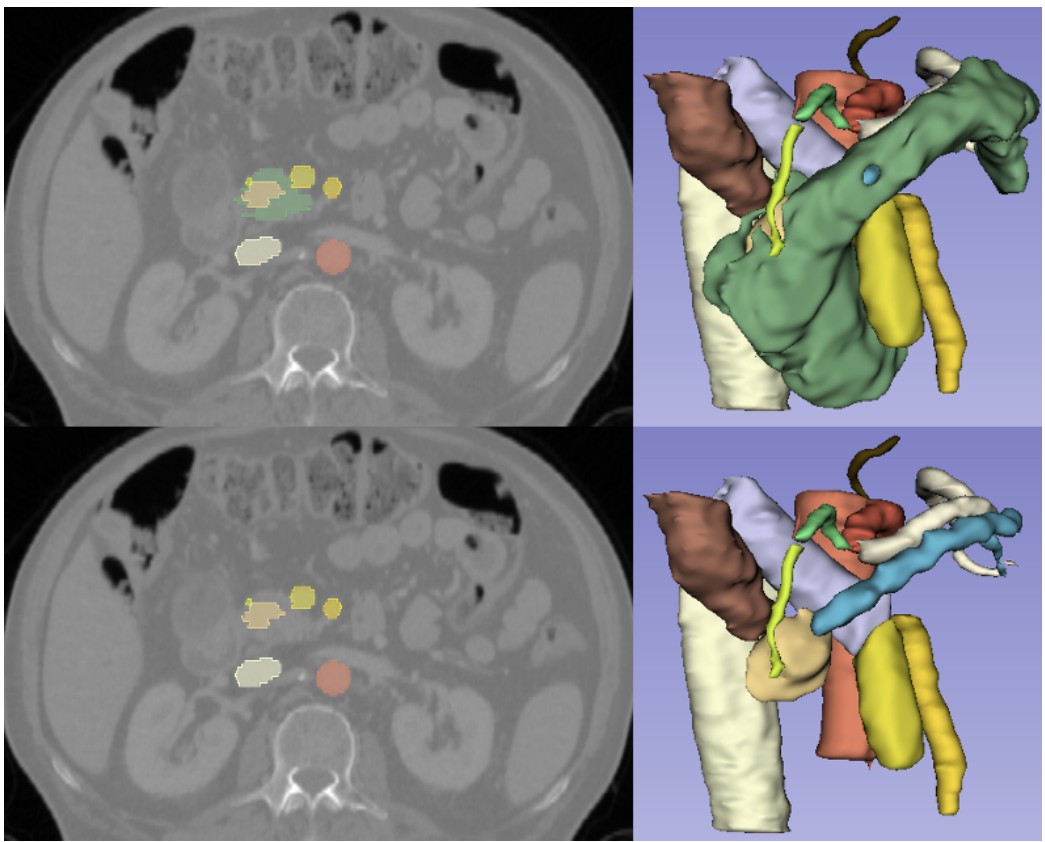

Figure 3: Qualitative examples of model predicted multiorgan and vascular segmentations. Left: axial CT slices with overlayed predicted labels. Right: corresponding 3D renderings illustrating pancreas, tumor, and major vessels.

