# OpenReview forum: "Multimodal Assessment of Pancreatic Cancer Resectability Using Deep Learning"
_MIDL.io/2026/Conference — MIDL 2026 Poster_

### Official Review · Reviewer_PMrJ · 2025-12-19

**Confidence:** 5
**Preliminary Rating:** 2
**Final Rating:** 3

**Summary:**

The paper presents a multimodal deep learning framework designed to classify Pancreatic Ductal Adenocarcinoma (PDAC) patients into three NCCN resectability categories: upfront resectable, borderline resectable, and locally advanced.

The architecture utilizes a Swin-UNETR backbone to extract anatomical features from 3D contrast-enhanced CT scans, which are then concatenated with an embedding of 17 structured clinical variables processed via an MLP.

A key methodological component is a dynamic multitask objective that adjusts the weighting between an auxiliary segmentation loss and the classification loss based on the tumor Dice score during training.

The approach is evaluated on a retrospective cohort of 159 patients using stratified nested 5-fold cross-validation, reporting an AUC of 0.86 and macro-F1 of 0.79.

**Strengths:**

The motivation to move away from purely geometric, segmentation-dependent inference toward an end-to-end classification system is clinically sound.

The integration of clinical variables reflects real-world tumor board decision-making.

The paper advances the state of the art by successfully implementing a multiclass NCCN prediction model that integrates non-imaging data.

The primary technical novelty is the dynamic multitask objective. Instead of fixed weights, the loss weights ($w_{seg}, w_{cls}$) adapt based on the running segmentation performance. This acts as a curriculum learning strategy. While effective, this is an incremental technical innovation rather than a paradigm shift.

The translation of voxel-level data into patient-level NCCN categories (including "borderline," which is notoriously difficult) is a valuable application.

**Weaknesses:**

The premise that "mask-free inference"  is a massive scalability advantage is slightly overstated. The model still requires a heavy 3D encoder (Swin-UNETR). The computational burden is similar to segmentation models, even if the mask isn't the final output.

While the architecture is standard, specific details on the "structured clinical variables" imputation (KNN k=5) are present, but the specific feature importance or correlation of these variables with the outcome is not analyzed. Code availability is not explicitly confirmed in the text provided.

The fusion mechanism (simple concatenation of feature vectors)  is technically trivial and lacks the sophistication of cross-attention mechanisms seen in top-tier multimodal literature.

The authors admit that more complex fusion mechanisms showed "no measurable benefit", which might suggest limitations in the dataset size rather than the efficacy of advanced fusion.

No external validation: The results are purely internal. In medical imaging, domain shift (different scanners, protocols) is a massive hurdle. Without external validation, the claim of a "robust and reproducible tool"  is unsupported

**Detailed Comments:**

Minor improvements can be done with improving figure quality and reducing the number of long sentences.

**Justification Of Final Rating:**

I intended to increase my score due to availability of the data/code (did not check fully), and also putting an external validation. Other questions are really arguable. The data size is small and real benefit is either unknown or marginal.

**Justification Of The Preliminary Rating:**

While the methodology is rigorous (nested CV) and the problem is clinically relevant, the work falls short of the high standards for top-tier venues  primarily due to data scale. An N of 159 with only ~27 examples of one class is insufficient to make robust claims about a deep learning model's efficacy, particularly one using a heavy backbone like Swin-UNETR. The technical novelty (dynamic loss weighting) is interesting but incremental. The paper is competent, likely suitable for a domain-specific conference (like MIDL), but viewed through the lens of a "high acceptance bar" reviewer, the lack of external validation and small sample size are fatal flaws.

**Questions To Address In The Rebuttal:**

please see weaknesses comments for self-describing questions to authors.
Furthermore, here are some further questions:

With only ~5 "locally advanced" cases in each test fold, how stable is the performance for this specific class? Please provide the standard deviation of the F1-score specifically for the "locally advanced" class across the folds.

You utilized KNN imputation for missing clinical values. What is the percentage of missing data? Did you analyze if the imputation introduces bias, particularly for highly predictive markers like CA 19-9?

---

> ### Author Response · Authors · 2026-01-23
> **Feedback to Reviewer PMrJ**
>
> 1) The premise that "mask-free inference" is a massive scalability advantage is slightly overstated. The model still requires a heavy 3D encoder (Swin-UNETR). The computational burden is similar to segmentation models, even if the mask isn't the final output.
>
> ANSWER)
> We thank the reviewer for this helpful comment. We agree that “scalability advantage” could be misinterpreted in terms of computational cost. Our intent was not to claim reduced inference-time complexity, as the model still relies on a full 3D Swin-UNETR encoder. Rather, mask-free inference refers to practical deployment: the model does not require segmentation masks as input at test time, eliminating the need for manual annotations or additional segmentation pipelines during clinical use. We have clarified this distinction in the revised manuscript and now emphasize workflow simplicity rather than computational savings (highlighted).
>
> 2) While the architecture is standard, specific details on the "structured clinical variables" imputation (KNN k=5) are present, but the specific feature importance or correlation of these variables with the outcome is not analyzed. Code availability is not explicitly confirmed in the text provided.
>
> ANSWER)
> We have now explicitly added a Code and Data Availability statement in the revised manuscript (highlighted in yellow), including a public repository link and a description of how de-identified data and trained model weights can be obtained for research purposes.
> Regarding analysis of the structured clinical variables, we agree that investigating variable level contributions can be informative. In this work, however, our primary goal was to develop and validate a multimodal end-to-end framework for NCCN resectability prediction rather than to perform a detailed clinical covariate association study.
> Importantly, we already provide systematic modality level attribution via ablation experiments (feature isolation, modality knockout, and single modality training), which directly quantify how much the clinical modality contributes to predictive performance in the presence of imaging. These experiments address the key question of whether structured clinical data adds measurable value beyond imaging alone.
> We note that variable level importance in the full multimodal setting is not straightforward to interpret, because the imaging branch produces a high dimensional learned representation and interactions between modalities can lead to unstable or cohort specific attributions. We therefore chose to keep the interpretability analysis focused on robust, model level ablations. We have clarified this scope in the revised manuscript and list variable level importance analysis as future work.
>
> 3) The fusion mechanism (simple concatenation of feature vectors) is technically trivial and lacks the sophistication of cross-attention mechanisms seen in top-tier multimodal literature.
> 4) The authors admit that more complex fusion mechanisms showed "no measurable benefit", which might suggest limitations in the dataset size rather than the efficacy of advanced fusion.
>
> ANSWER)
> We thank the reviewer for this comment. While more sophisticated fusion mechanisms such as cross-attention can be beneficial in large scale multimodal settings, our preliminary experiments showed no performance gain on this dataset and reduced training stability compared to simple concatenation. We attribute this to the limited cohort size, as cross-attention introduces many additional parameters and is data hungry. In contrast, late fusion via concatenation provided the most robust and well regularized integration, as confirmed by both internal cross-validation and external validation. We have clarified this empirical design choice in the revised manuscript (highlighted in yellow).
>
> 5) No external validation: The results are purely internal. In medical imaging, domain shift (different scanners, protocols) is a massive hurdle. Without external validation, the claim of a "robust and reproducible tool" is unsupported
>
> ANSWER)
> We thank the reviewer for emphasizing the importance of external validation, particularly given domain shift in medical imaging. In response, we have added an external validation on an independent cohort from Kantonsspital Aarau (KSA Aarau), which was not used for training, tuning, or internal cross-validation. On this dataset, the model achieved an AUC of 0.86, a macro-F1 of 0.81, and an accuracy of 0.87, closely matching internal performance. We have updated the manuscript to include this validation, report the results in a dedicated table, and revise robustness claims accordingly (highlighted in yellow).

---

> > ### Author Response · Authors · 2026-01-23
> > **Further Feedback to Reviewer PMrJ**
> >
> > 6) With only ~5 "locally advanced" cases in each test fold, how stable is the performance for this specific class? Please provide the standard deviation of the F1-score specifically for the "locally advanced" class across the folds.
> >
> > ANSWER)
> > We thank the reviewer for this important point. The original submission reported macro-averaged F1 scores, which do not reflect class-specific variability. Given that the locally advanced class is the smallest, we have now added per-class F1-scores across the five outer folds and report the mean ± standard deviation for this class in the Appendix (Table 6). Although variability is higher, performance remains stable across folds. External validation on the independent KSA Aarau cohort further supports robustness beyond internal cross-validation.
> >
> > 7) You utilized KNN imputation for missing clinical values. What is the percentage of missing data? Did you analyze if the imputation introduces bias, particularly for highly predictive markers like CA 19-9?
> >
> > ANSWER)
> > We thank the reviewer for raising this important point regarding missing data handling and potential imputation bias, particularly for clinically relevant markers such as CA 19-9.
> > In the Basel cohort, missing values were present only for a subset of clinical variables and affected a small fraction of entries; we have now explicitly reported the percentage of missing values per variable in the revised manuscript. Continuous variables were imputed using KNN imputation (k=5), while categorical variables were imputed using the mode, applied strictly within the training folds to avoid data leakage.
> > To assess whether this imputation strategy introduced bias, we leveraged the newly added external validation cohort from Kantonsspital Aarau (KSA Aarau). Importantly, this external dataset contained complete clinical information for the variables used in our model, including CA 19-9, and therefore did not require imputation. Despite this difference, the model achieved performance comparable to the internal cross-validation results (AUC 0.86, macro-F1 0.81, accuracy 0.87).
> > The consistent performance on a dataset without imputed clinical values suggests that the KNN-based imputation did not introduce a systematic bias that the model relied upon, at least for the variables and cohorts considered in this study. We have clarified this point in the revised manuscript and explicitly discussed imputation as a potential limitation, noting that further validation on additional external cohorts would be valuable to confirm this finding.
> >
> > In case of more comments or questions we are more than happy to discuss these in the discussion phase

---

> > > ### Comment · Reviewer_PMrJ · 2026-02-01
> > > **not all questions answered but good enough**
> > >
> > > I intended to increase my score due to availability of the data/code (did not check fully), and also putting an external validation. Other questions are really arguable. The data size is small and real benefit is either unknown or marginal.

---

### Official Review · Reviewer_jk9v · 2026-01-08

**Confidence:** 5
**Preliminary Rating:** 5
**Final Rating:** 5

**Summary:**

The authors present an end-to-end multimodal model that combines 3D CT imaging with clinical data for accurate determination of pancreatic ductal adenocarcinoma (PDAC) resectability. The imaging modality is processed through a Swin-UNETR based segmentation model, and the extracted features are then fused with clinical embeddings via a classification head. The workflow is justified with elements of sound analysis alongside specification of the different types of loss. The authors produce a fusion based on a 288 combinatory vector which has been generated via 256 embedding of the image, and a 32-dimensional clinical embedding. The authors report results on 159 patients, evaluating the methodology using Dice scores for segmentation and standard classification metrics including AUC, Macro-F1, and accuracy. The results are compared with state-of-the-art prior work whilst including comparison between the multi-modal approach with single modality pipelines, achieving an AUC of 0.86 and outperforming the mentioned baseline methods.

**Strengths:**

- The paper discusses an important topic related to one of the most lethal malignancies and provides a sound and well-justified motivation for the problem. The gaps are clearly discussed, and the associated limitations and contributions are presented in a way that allows the reader to follow the work easily.
- The related work begins with a discussion of AI/ML approaches, providing insight into existing gaps and how prior studies have addressed the problem. The analysis is fair and balanced, highlighting both positive and negative aspects.
- The paper is easy to follow, and sufficient explanations and details are provided regarding the methodology, ensuring reproducibility and repeatability.
- The results are compared not only with uni-modal settings but also with state-of-the-art methods, providing an in-depth understanding of the impact of the proposed methodology.

**Weaknesses:**

In some parts of the paper, a few minor details are required to fully understand the underlying settings; however, these are minor issues and clarifying them would simply help readers gain a deeper understanding.

**Detailed Comments:**

There are only a few minor points that would benefit from clarification:

Section 3: Method
- In the segmentation Masks subsection, the authors mention that the segmentation loss emphasizes tumor and major vessels (SMA, SMV, portal vein, celiac trunk) as the Dice and Focal components give higher relative importance to small, clinically relevant structures. It would be helpful if the authors describe how this design choice impacts the downstream tasks and feature merging.

- In the Clinical Data subsection, the authors mention 17 variables. Are these all the clinical variables associated with the disease or were they selected based on a specific criterion or method? In addition, it would be helpful if the authors spcify the type of the clinical data and whether any text-to-numerical conversions were required. It also appears that the authors converted the 17 clinical variables into a 32-length vector using a single hidden layer MLP. What is the justification for this dimensional expansion?

- With regard to the model’s architecture, the authors mention adopting the Swin-UNETR transformer to extract local and global features. Is the reported 256-dimensional representation derived from global features, local features or merged embedding of both? Clarification on this point would be helpful.


Section 4: Experiments and Results

- The authors mention that the data was collected from two hospitals. Were there any challenges related to data normalization, acquisition settings, or data integration? An outline in this regard would be helpful for understanding the dataset in terms of reproducibility and for informing future studies.

**Justification Of Final Rating:**

Thank you for the detailed explanations, additional analysis and experimentation. The responses were in depth and the revisions are promising, increasing the technical transparency. The proposed approach is promising and has a solid foundation for future research in this field. My final rating remains strong accept.

**Justification Of The Preliminary Rating:**

I believe this paper receives a strong accept due to it addressing an important and clinically relevant problem with a well-motivated and carefully designed multi-modal approach. The methodology is clearly described, technically sound, and sufficiently detailed to support reproducibility. The experimental evaluation is thorough, including comparisons against uni-modal baselines and state-of-the-art methods, and the results consistently demonstrate meaningful performance improvements. Overall, the paper is well written, the contributions are clearly articulated, and the work represents a strong and complete study that would be valuable from both a clinical and AI point of view.

**Questions To Address In The Rebuttal:**

I do not have any further questions for the rebuttal phase.

---

> ### Author Response · Authors · 2026-01-23
> **Feedback to Reviewer jk9v**
>
> We sincerely thank Reviewer 2 for the strong accept and for the constructive and encouraging feedback. We are pleased that the reviewer finds the proposed approach relevant and well motivated. Below, we address the specific comment in detail.
>
>
> 1) In the segmentation Masks subsection, the authors mention that the segmentation loss emphasizes tumor and major vessels (SMA, SMV, portal vein, celiac trunk) as the Dice and Focal components give higher relative importance to small, clinically relevant structures. It would be helpful if the authors describe how this design choice impacts the downstream tasks and feature merging.
>
> ANSWER)
> We thank the reviewer for this insightful comment. Emphasizing tumor and major vessels in the segmentation loss encourages the shared encoder to learn fine grained, vessel-aware anatomical features that are directly propagated to the classification branch. Because the segmentation decoder and classification head share the same encoder, these anatomy aware features form the 256-dimensional imaging embedding used for multimodal fusion with clinical data.
>
> This design stabilizes feature fusion and improves resectability classification, as reflected by higher macro-F1 and AUC compared to unimodal and segmentation only baselines, and is further supported by our ablation studies. We have clarified this interaction in the revised manuscript (highlighted).
>
> 2)
> In the Clinical Data subsection, the authors mention 17 variables. Are these all the clinical variables associated with the disease or were they selected based on a specific criterion or method? In addition, it would be helpful if the authors spcify the type of the clinical data and whether any text-to-numerical conversions were required. It also appears that the authors converted the 17 clinical variables into a 32-length vector using a single hidden layer MLP. What is the justification for this dimensional expansion?
>
> ANSWER)
> We thank the reviewer for this thoughtful comment. The 17 clinical variables were preselected a priori by experienced physicians during multidisciplinary tumor board discussions and reflect routinely available factors relevant for PDAC operability assessment rather than an exhaustive disease feature set. All variables are structured tabular data; continuous variables were standardized and categorical variables were numerically encoded, with no free-text data requiring text-to-numeric conversion.
>
> The clinical variables were embedded into a 32-dimensional representation using a lightweight MLP, selected empirically (powers of two) to balance expressive capacity and regularization and to yield a compact latent space compatible with the 256-dimensional imaging embedding for multimodal fusion. We have clarified these points in the revised manuscript (highlighted).
>
> 3)
> With regard to the model’s architecture, the authors mention adopting the Swin-UNETR transformer to extract local and global features. Is the reported 256-dimensional representation derived from global features, local features or merged embedding of both? Clarification on this point would be helpful.
>
> ANSWER)
> The 256-dimensional imaging representation corresponds to the Swin-UNETR encoder bottleneck and represents a merged embedding of local and global features learned via hierarchical, shifted-window self-attention. This bottleneck integrates fine grained vessel tumor interactions with global anatomical context and is used as the shared imaging embedding for both segmentation and multimodal fusion. We have clarified this in the revised manuscript (highlighted).
>
> 4)
> The authors mention that the data was collected from two hospitals. Were there any challenges related to data normalization, acquisition settings, or data integration? An outline in this regard would be helpful for understanding the dataset in terms of reproducibility and for informing future studies.
>
> ANSWER)
> Thank you. The data were collected retrospectively from two institutions using standard-of-care contrast-enhanced CT protocols. While scanner vendors and acquisition settings varied across sites, no major challenges arose that required site-specific preprocessing or harmonization.
> To ensure robustness and reproducibility across institutions, all CT volumes were resampled to a uniform isotropic resolution, intensity normalized, and processed using an identical preprocessing pipeline. Additionally, extensive spatial and intensity augmentations were applied during training to improve robustness to acquisition variability. Clinical variables were harmonized using consistent definitions and units across sites prior to modeling.
> We have added a brief clarification in the revised manuscript describing the data integration and normalization steps used to mitigate inter-site variability (highlighted in yellow).
>
> In case of more comments or questions we are more than happy to discuss these in the discussion phase

---

### Official Review · Reviewer_jXPc · 2026-01-08

**Confidence:** 5
**Preliminary Rating:** 4
**Final Rating:** 5

**Summary:**

In this study, the authors describe a deep learning approach that focuses on accurate determination of pancreatic ductal adenocarcinoma (PDAC) resectability. The authors aligned well with current clinical guidelines and focused to predict the 3 classes (upfront resectable, borderline resectable,
locally advanced) defined by the NCCN guidelines. The deep learning approach uses state of the art methodology and the auhors have added ablation studies to show the effect of the different components and the effect of their multimodal approach. The dataset is coming from two Swiss hospitals and is reasonable, but not large: a total of 159 patients. Experiments and results are based on 5-fold CV on this full dataset.

**Strengths:**

- Strong methodology with good accompanying ablation experiments.
- Good alignment with clinical guidelines
- Clear rationale and explanation of potential clinical benefit.
- Multimodal, so taking clinical features into account - very important.

**Weaknesses:**

- Weak reference standard: the authors explain in their introduction that there is substantial interobserver variability for determining the NCCN category, but it seems the judgement from just one reader - the reader that did the NCCN resectability during clinical practice - was used as the reference. Wouldn't it be better to make a reference standard with multiple readers.
- In relation to my first point: there is no comparison to current performance of one clinician. Is this AI system better or worse than current clinical practice? If it is worse, how would it help? A comparison with a second human observer would really have helped. If there would have been a reference standard set by multiple clinical readers, the authors could have measured the performance of this AI system, and potentially of another independent single human observer.
- No external validation.
- No reference to data, code or model weights availability for future research.

**Detailed Comments:**

- The authors explain that missing variables were imputed. Please add how often this was necessary. What percentage of values was missing for the clinical variables?
- Please clarify on what data the ablation experiments were performed. The full dataset? In the manuscript, it reads "We use simple concatenation for fusion, as ablation experiments showed no measurable benefit from more complex cross attention mechanisms." So, if all the ablation experiments are also conducted on the full dataset using 5-fold CV, isn't there a potential bias?

**Justification Of Final Rating:**

I would like to thank the authors for their rebuttal. Excellent explanations, and I want to be compliment the authors on extending the work with an additional external dataset. I have adapted my score to Strong accept.

**Justification Of The Preliminary Rating:**

- Clearly written paper with extensive results
- Clinically relevant problem, and paper is well aligned with the clinical guidelines.
- State of the art methodology combined with appropriate ablation experiments.

**Questions To Address In The Rebuttal:**

- Please address my comments about the reference standard.
- Please add sentences about code, weights and data availability.

---

> ### Author Response · Authors · 2026-01-23
> **Feedback to Reviewer jXPc**
>
> Dear Reviewer, we thank you for your valuable input. With your feedback, we can increase the overall quality of the work.
> Below you will find the answers to each of your points you have mentioned. In addition, we will upload a revised manuscript with your recommendations implemented and highlighted in yellow.
>
> Weaknesses:
> 1) Weak reference standard: the authors explain in their introduction that there is substantial interobserver variability for determining the NCCN category, but it seems the judgement from just one reader - the reader that did the NCCN resectability during clinical practice - was used as the reference. Wouldn't it be better to make a reference standard with multiple readers.
>
> ANSWER)
> We thank the reviewer for this important point. The NCCN resectability labels were derived from multidisciplinary tumor board (MDT) decisions during routine clinical practice, involving physicians from surgery, radiology, and oncology, and represent the clinical reference standard at both institutions. To further ensure consistency, all MDT-based labels were independently reviewed by an additional physician not involved in model development. Segmentation masks used for auxiliary supervision were generated and quality controlled in a multi-expert workflow, with annotations reviewed by up to five physicians. We have clarified this reference standard and annotation process in the revised manuscript (highlighted).
>
> 2) In relation to my first point: there is no comparison to current performance of one clinician. Is this AI system better or worse than current clinical practice? If it is worse, how would it help? A comparison with a second human observer would really have helped. If there would have been a reference standard set by multiple clinical readers, the authors could have measured the performance of this AI system, and potentially of another independent single human observer.
>
> ANSWER)
> We thank the reviewer for this important suggestion. A direct clinician AI comparison would indeed be valuable, but it was not the objective of this retrospective study. Our model was trained and evaluated against the clinical reference standard used in practice, namely MDT consensus resectability assessment, which already integrates input from multiple physicians. A fair clinician AI comparison would require a dedicated prospective multi-reader study with standardized reading conditions, which was beyond the scope of the available dataset. We do not claim superiority over clinicians; rather, we position the model as a decision support tool to provide standardized, reproducible NCCN-consistent predictions for MDT discussions. This has been clarified in the revised manuscript (highlighted). But its an interesting idea and we will tackle that in a follow up project.
>
> 3) No external validation.
>
> ANSWER)
> We agree that cross-institution robustness is critical in medical imaging due to potential domain shift. In response, we added an external validation using an independent cohort from Kantonsspital Aarau (KSA Aarau), which was not used for model training, tuning, or internal cross validation. The model trained on the Basel cohort was evaluated directly on the KSA Aarau data without retraining and achieved performance comparable to internal results, supporting cross-institution generalization. The external validation setup and results are now described in a dedicated subsection in the revised manuscript (highlighted in yellow).
>
> 4) No reference to data, code or model weights availability for future research.
>
> ANSWER)
> Thank you for that comment. We have specified the reproducibility statement in the submission tool and provided the GitHub repository link. However, we agree that the original submission did not sufficiently specify data and code availability.
> In response, we have now explicitly added a Data and Code Availability statement to the revised manuscript. The full training and evaluation code is publicly available at
> https://github.com/vincentochs/pancreas_resectability.
> Due to patient privacy regulations and institutional review board (IRB) constraints, the raw imaging and clinical data cannot be released publicly. However, we clarify that the data will be made available by the corresponding author upon reasonable request, subject to appropriate approvals.
> In addition, we have added a brief statement in the abstract to clearly communicate code availability. All changes are highlighted in yellow in the revised manuscript.
>
> 5) Please address my comments about the reference standard.
>
> ANSWER)
> We did - please look above.
>
> 6) Please add sentences about code, weights and data availability.
>
> ANSWER)
> We did - please have a look above.
>
>
> In case of more comments or questions we are more than happy to discuss these in the discussion phase
> Once again, thank you for your comments!

---

### Author Rebuttal · Authors · 2026-01-23

**Rebuttal:**

General Rebuttal Statement

We sincerely thank all reviewers for their careful evaluation of our manuscript and for their constructive, detailed feedback. We appreciate the time and expertise invested in reviewing our work, which has substantially helped to improve the clarity, rigor, and completeness of the paper.

All reviewer comments have been carefully addressed in the revised manuscript. In response, we have:

-clarified the reference standard and annotation workflow, explicitly describing the multidisciplinary tumor board (MDT) based labeling and multi-physician validation process;

-added a new external validation on an independent cohort from Kantonsspital Aarau (KSA Aarau), including quantitative results, to demonstrate robustness to inter institutional domain shift;

-expanded methodological details on clinical variable selection, missingness, imputation strategy, and reported missingness percentages;

-clarified architectural choices, including feature representations, fusion strategy, and the rationale for simple concatenation over cross-attention;

-added per-class performance statistics, including class-wise F1-score variability;

-explicitly stated code, model weight, and data availability;

-refined claims regarding mask-free inference to emphasize clinical applicability rather than computational complexity; and

-extended the limitations and discussion sections to transparently address remaining constraints and outline future work.



All additions and modifications are clearly highlighted in the revised manuscript to facilitate review. We believe these revisions substantially strengthen the manuscript and better align it with the expectations of the community.

We thank the reviewers again for their valuable insights and for helping us improve the quality and impact of this work.

**Supporting Material:**

/attachment/ad0b4347b53a53df75f68592fd230f4db78d85b8.pdf

---

### Author Response · Authors · 2026-01-29
**Discussion Phase**

Following up on our rebuttal - I am happy to answer any remaining questions.

Thank you

---

### Meta-Review · Area_Chair_5XVx · 2026-02-09

**Recommendation:** Accept (Oral)
**Confidence:** 5

**Metareview:**

Two reviewers recommend strong accept and one reviewer recommends borderline.  Overall, it is a technically sound method with sufficient  experimental evaluation, though the small dataset is a limiting factor.

---

### Decision · Program_Chairs · 2026-02-13

Accept (Poster)